# What We Are Learning from the Diverse Structures of the Homodimeric Type I Reaction Center-Photosystems of Anoxygenic Phototropic Bacteria

**DOI:** 10.3390/biom14030311

**Published:** 2024-03-06

**Authors:** Robert A. Niederman

**Affiliations:** Department of Molecular Biology and Biochemistry and Rutgers Climate and Energy Institute, Rutgers University, Piscataway, NJ 08854-8082, USA; rniederm@dls.rutgers.edu; Tel.: +1-917-783-9386

**Keywords:** chloracetobacteria, green sulfur bacteria, heliobacteria, light-harvesting proteins, molecular evolution, photosystems, protein structure, reaction centers

## Abstract

A Type I reaction center (RC) (Fe-S type, ferredoxin reducing) is found in several phyla containing anoxygenic phototrophic bacteria. These include the heliobacteria (HB), the green sulfur bacteria (GSB), and the chloracidobacteria (CB), for which high-resolution homodimeric RC-photosystem (PS) structures have recently appeared. The 2.2-Å X-ray structure of the RC-PS of *Heliomicrobium modesticaldum* revealed that the core PshA apoprotein (PshA-1 and PshA-2 homodimeric pair) exhibits a structurally conserved PSI arrangement comprising five C-terminal transmembrane α-helices (TMHs) forming the RC domain and six N-terminal TMHs coordinating the light-harvesting (LH) pigments. The *Hmi. modesticaldum* structure lacked quinone molecules, indicating that electrons were transferred directly from the A_0_ (8^1^-OH-chlorophyll (Chl) *a*) acceptor to the F_X_ [4Fe-4S] component, serving as the terminal RC acceptor. A pair of additional TMHs designated as Psh X were also found that function as a low-energy antenna. The 2.5-Å resolution cryo-electron microscopy (cryo-EM) structure for the RC-PS of the green sulfur bacterium *Chlorobaculum tepidum* included a pair of Fenna–Matthews–Olson protein (FMO) antennae, which transfer excitations from the chlorosomes to the RC-PS (PscA-1 and PscA-2) core. A pair of cytochromes *c_Z_* (PscC) molecules was also revealed, acting as electron donors to the RC bacteriochlorophyll (BChl) *a*’ special pair, as well as PscB, housing the [4Fe-4S] cluster F_A_ and F_B_, and the associated PscD protein. While the FMO components were missing from the 2.6-Å cryo-EM structure of the Zn- (BChl) *a*’ special pair containing RC-PS of *Chloracidobacterium thermophilum*, a unique architecture was revealed that besides the (PscA)_2_ core, consisted of seven additional subunits including PscZ in place of PscD, the PscX and PscY cytochrome *c* serial electron donors and four low mol. wt. subunits of unknown function. Overall, these diverse structures have revealed that (i) the HB RC-PS is the simplest light–energy transducing complex yet isolated and represents the closest known homolog to a common homodimeric RC-PS ancestor; (ii) the symmetrically localized Ca^2+^-binding sites found in each of the Type I homodimeric RC-PS structures likely gave rise to the analogously positioned Mn_4_CaO_5_ cluster of the PSII RC and the Tyr_Z_ RC donor site; (iii) a close relationship between the GSB RC-PS and the PSII Chl proteins (CP)43 and CP47 was demonstrated by their strongly conserved LH-(B)Chl localizations; (iv) LH-BChls of the GSB-RC-PS are also localized in the conserved RC-associated positions of the PSII Chl_Z-D1_ and Chl_Z-D2_ sites; (v) glycosylated carotenoids of the GSB RC-PS are located in the homologous carotenoid-containing positions of PSII, reflecting an O_2_-tolerance mechanism capable of sustaining early stages in the evolution of oxygenic photosynthesis. In addition to the close relationships found between the homodimeric RC-PS and PSII, duplication of the gene encoding the ancestral Type I RC apoprotein, followed by genetic divergence, may well account for the appearance of the heterodimeric Type I and Type II RCs of the extant oxygenic phototrophs. Accordingly, the long-held view that PSII arose from the anoxygenic Type II RC is now found to be contrary to the new evidence provided by Type I RC-PS homodimer structures, indicating that the evolutionary origins of anoxygenic Type II RCs, along with their distinct antenna rings are likely to have been preceded by the events that gave rise to their oxygenic counterparts.

## 1. Introduction

In contrast to the heterodimeric Type II reaction centers (RCs) (pheophytin–quinone (Q) type, Q reducing) of the well-described anoxygenic purple chlorophototrophic (i.e., forming chlorophylls Chls) bacteria, homodimeric Type I RCs ([4Fe-4S] cluster type, ferredoxin reducing) are found in the heliobacteria (HB), green sulfur bacteria (GSB), and chloracidobacteria (CB) [1], where they also catalyze the primary photochemical reactions. These anoxygenic Type I structures are believed to have arisen from the primordial homodimeric RC (the last common RC ancestor) that subsequently gave rise to both the extant heterodimeric photosystem (PS) I and PSII RCs of the oxygenic phototrophs [2], which together catalyze the transmembrane charge separations resulting in the respective reduction of [4Fe-4S] centers and Q molecules, as accompanied in PSII by O_2_ evolution. The individual Type I subunits forming the cores of the homodimeric RC-PSs complexes [3,4,5,6], designated as PshA in HB [3] and as PscA in GSB [4,5] and CB [6], contain eleven transmembrane α-helices (TMHs), with the six N-terminal helices binding the light-harvesting (LH) bacteriochlorophylls (BChls) that transfer excitation energy to the RC cofactors housed within the five C-terminal helices. This structural organization is conserved in the multifunctional PsaA and PsaB proteins of PSI, which also possess fused antenna and RC domains [7]. By contrast, the structurally conserved Type II RCs of the purple bacteria are surrounded by unique elliptical assemblies of LH1 proteins consisting of up to 17 subunits to form RC-LH1 complexes (the structures of these RC-LH1 complexes have been recently reviewed in detail [8,9,10,11]).

The electron transfer cofactors of the homodimeric Type I RCs include the P800/P840 special pair BChls, two accessory (B)Chls (A_CC_), and two Chl (A_0_) acceptors, which, together with the [4Fe-4S] cluster (F_X_), form a bifurcated chain capable of functioning in the absence of the A_1_ menaquinone (MQ) component that transfers electrons between the A_0_ and F_X_ electron carriers of PSI [1]. The light-driven charge separation created by the homodimeric Type I RC powers cyclic electron transfer pathways, creating a transmembrane electrochemical proton gradient that drives the synthesis of adenosine 5′-triphosphate. The remaining BChl molecules, which include 56, 26, and 24 BChls in the HB, GSB, and CB, respectively [3,4,5,6], function in LH capacities, with the additional HB BChls representing an expanded PshA-associated antenna, in part compensating for the lack of chlorosomes and the Fenna–Matthews–Olson protein (FMO) antennae present in the GSB and CB.

In this review, I will describe the recently elucidated high-resolution structures of the homodimeric Type I RC-PSs from HB [3], GSB [4,5], and CB [6], with emphasis on their structural diversity and the manner in which they provide an improved understanding of both the LH and primary photochemical events catalyzed by these unique pigment–protein complexes. Moreover, I will detail how these structures have contributed to our current understanding of the evolutionary origins of RC-PSs, especially with regard to the relationships that the observed homodimeric architectures have to the emergence of the extant heterodimeric PSI and PSII complexes of the oxygenic phototrophs.

## 2. Type I Homodimeric Reaction Center—Photosystems of Anoxygenic Phototrophs

Unlike higher phototrophs, in which separate *psaA* and *psaB* genes encode the PsaA and PsaB apoproteins that form heterodimeric Type I RCs, a single *pshA* gene encodes the HB RC-PS homodimer, while the RC-PSs of GSB and CB are encoded by single *pscA* genes. These unique homodimeric RCs apparently arose from the last common PSI ancestor that served as the evolutionary precursor of the heterodimeric Type I RC [12] in which duplication of the gene encoding an ancestral core homodimeric RC apoprotein, followed by genetic divergence likely accounts for the appearance of the Type I RC in their heterodimeric form [13]. While high-resolution structures have long been available for the heterodimeric RCs of the purple bacteria [14,15] as well as for both PSI [7] and PSII [16], elucidation of the structures of Type I RC-PS complexes of the HB, GSB, and CB have proved to be of considerable importance in obtaining a more complete picture of these early photosynthetic pigment–proteins, with their apparent close relationships to the ancestral RC protein.

### 2.1. Structure of the Reaction Center-Photosystem of the Heliobacteria

The earliest structure of Type I homodimeric RC-PS was determined for *Heliomicrobium modesticaldum* [3], a thermophilic anaerobe isolated from hot spring and volcanic soil environments [17]. *Hmi. modesticaldum*, as a member of the phylum Firmicutes, is among the simplest known complexes capable of catalyzing primary photochemical reactions. In addition, this RC-PS is considered most closely related to the common RC-PS progenitor [3,12]. Figure 1A,B display the 2.2-Å resolution X-ray crystallography structure determined for the *Hmi. modesticaldum* RC-PS by Gisriel et al. [3]. The structure exhibits perfect C_2_ symmetry, suggesting that it is likely a close homolog of the homodimeric ancestral RC, which is also supported by the anoxic niches in which HB are found, comparable to those of the early earth. This anoxygenic phototroph is further unique in lacking the peripheral antenna components found in the GSB and CB, as well as in containing an RC-PS largely utilizing BChl *g*, an isomer of Chl *a* [18].

The overall molecular architecture is structurally homologous to known Type I RCs, as confirmed in Figure 1B,C, which shows that the PshA subunits form eleven TMHs, with the first six containing the antenna BChls and the remaining five comprising the RC domain. Two additional α-helices labeled “X” are also found, located in approximately the same positions as the PsaI and PsaJ proteins in the heterodimeric PSI. Recent spectroscopic evidence demonstrated that protein X, which binds two BChl *g* molecules, serves as a low-energy LH component involved in the uphill transfer of excitations to the RC [19]. While both protein X and PsaJ also bind antenna (B)Chl pigments, their lack of significant sequence identity (SI) suggests the common position occupied by these subunits reflects convergent evolution. A total of 54 BChl *g* function as LH-BChls, while four BChl *g*’, two 8^1^-hydroxy-Chl *a*, two 4,4′-di-aponeurosporenes, two lipids, and one [4Fe-4S] cluster are also associated with the *Hmi. modesticaldum* RC-PS.

The *C*_2_ symmetry axis, assuming cofactors of the RC electron transfer chain, gives rise to two essentially identical cofactor branches, as governed by the homodimeric nature of the HB RC (Figure 2A). Residues coordinating the cofactors consist of His 537 for the BChl *g*’ dimer special pair, together with Cys432 and Cys441 for the F_X_ terminal acceptor as provided by the individual PshA subunits which give rise to the symmetrical arrangement. However, the A_0_ 8^1^-OH-Chl *a* primary acceptor and the A_CC_ accessory Bchl *g*’ molecules are both likely coordinated by H_2_O molecules and an unidentified molecule of a size similar to a chloride ion, respectively, which are, in turn, H-bonded to the respective Ser545 and Gln458 side chains. It should be noted that recent 2-D hyperfine sublevel correlation (HYSCORE) spectroscopy and density function theory (DFT) studies of PSI [20] demonstrated that Chl_2A_ and Chl_3A_ located in the respective positions of the HB A_CC_ and A_0_, form a Chl dimer that functions as the primary electron acceptor in which extensive delocalization promotes a prolonged charge separation, resulting in the high quantum efficiency of PSI. This mechanism may apply to the HB RC since ACC (BChl *g*) and A_0_ (8^1^-OH-Chl *a*) possess the conserved orientation similar to the homologous PS I cofactors and the ~9.0 Å Mg → Mg distances between them [3,7].

Up to two MQ molecules have been found in purified HB RCs [22] despite the absence of a conserved Q binding site. The MQ molecules are loosely bound and were lost during the purification of the crystalline RC-PS [3]. Recently, a model was constructed by Kashey et al. [21] (Figure 2B) in which MQ molecules were placed in potential locations between A_0_ and F_X_. This was based upon unassigned electron density in the crystal structure sufficient for placement of an isoprenyl tail [3] (Figure 2C). A comparison of the phylloquinone (PQ) binding site of PSI with the potential MQ site for PshA (Figure 2D) revealed conservation of only a Leu residue homologous to the Leu722 of PsaA on one side of the PQ head group, while the Trp697 on the opposite side and the nearby Phe689 are replaced by Arg554 and Met546 residues, respectively, in the HB RC. Since the distance between A_0_ and F_X_ is shorter than in PSI, where the electron carrier PQ is positioned between them, electron transfer in the HB is capable of occurring between A_0_ and F_X_ at an intrinsic slowed rate without the need for an intermediate Q. Nevertheless, as a lipophilic molecule, MQ was shown to function as an alternative electron carrier under high-light conditions, in an analogous manner to mobile Q molecules in Type II RCs, when the final soluble ferredoxin acceptor pool is replete [21]. It is thought that under these circumstances, electrons are instead transferred directly to the cytochrome *b*_6_*c* complex rather than via the ferredoxin/NADP^+^ oxidoreductase and NAD(P)H dehydrogenase as is the case with ferredoxin. This ability to alternatively engage MQ as the terminal RC electron acceptor points to the inadequacy in the classification of the HB RC as solely belonging to Type I [12].

The loosely bound PshB subunit lost during purification of the HB RC-PS core complex is considered to function as a ferredoxin electron acceptor [23] rather than as the bound F_A_ and F_B_ terminal electron acceptors of the PsaC subunit of PSI, thus requiring that F_X_ function in this role [23]. Patches of positive charge on the cytoplasmic surface of the HB RC provided by Lys584 and Lys587 of PshA, as well as twin Lys423 residues extending out on either side of the F_X_ site, have been modeled as a potential interaction site for the negatively charged surface of PshB [3].

It is generally accepted that photochemical RCs arose only once in the form of a homodimeric complex more than 3 billion years ago and ultimately evolved into currently existing heterodimeric versions. While the homodimeric *Hmi. modesticaldum* PS represents the simplest such complex isolated thus far, it is not thought to signify the actual ancestral form but rather a close homolog of the Type I RC common ancestor [24]. This is based on the likelihood that the presence of the RC-PS protein in *Hmi. modesticaldum* originated from horizontal gene transfer, a possibility supported by the presence of *pshA* in a single gene cluster along with the accompanying pigment synthesis genes [25].

A comparison of structural details from the homodimeric HB RC with all known RC structures has led to proposals on the structural and functional properties of a common RC ancestor as they relate to the evolutionary origin of the resulting Type I and Type II complexes [12,13]. Emphasis was placed upon the loose Q-binding capacity of the HB RC, thought also to be a property of the homodimeric ancestral complex, purported to bind two mobile Q molecules participating in Q reduction via an inefficient disproportionation reaction involving simultaneous oxidation and reduction in Q redox species. Optimization of this reaction via subsequent evolutionary changes led to a divergence between the two types of extant RCs. For Type I RCs, a [4Fe-4S] cluster was added that enables double Q reduction, while heterodimerization led to affixing a Q molecule into the Q_A_ site on the D1 subunit of PSII within electron transferring distance from the second (Q_B_) binding site on D2, where a gated double Q reduction to a hydroquinone is catalyzed via a radical anion Q intermediate in Type II RC. In Type I RCs, subsequent fixation of the Q sites was followed by heterodimerization to permit the association of the PsaC subunit housing the F_A_ and F_B_ [4Fe-4S] clusters in response to O_2_ levels arising after the oxygen-evolving complex appeared. Further possible evolutionary scenarios will be discussed in Section 3, following the presentation of the GSB and CB RC-PS structures (Section 2.2 and Section 2.3, respectively).

### 2.2. Structure of the Reaction Center-Photosystem of the Green Sulfur Bacteria

The green sulfur bacterium *Chlorobaculum tepidum*, like *Hmi. modesticaldum*, is a thermophilic obligate anaerobe that also contains a Type I homodimeric RC with the RC protein (PscA)_2_ as a core. The LH capabilities of *Cba. tepidum* is considerably enhanced by a chlorosome peripheral antenna system containing self-assembled Chl *c* molecules that exist as a lamellar structure lacking a protein scaffold [26]. The chlorosomes are connected by a baseplate to two copies of the trimeric FMO, in which each protomer contains eight BChls *a,* from which excitons are passed onto the RC-core of the RC-PS complex. While energy transfer efficiencies between the Chls *c* within chlorosomes to the FMO BChls *a* and within FMO approach the 100% levels found for transfer from LH Chls to the PSI RC, the efficiencies in the GSB range from only 35 to 75% for the FMO trimer to the RC-PS [27,28,29,30]. In an effort to elucidate the basis for the reduction in energy transfer efficiency and to obtain an improved understanding of the overall excitation energy and electron transfer pathways, a complete 2.5 Å cryo-electron microscopy (EM) structure of the FMO-containing *Cba. tepidum* RC-PS was recently determined by Xie et al. [4] (Figure 3A,B). This was accomplished with a complex purified after solubilization with n-dodecly-β-D-maltoside and detergent exchange to lauryl maltoside neopentylglycol; the resulting structural model follows an earlier 2.7 Å cryo-EM structure [5] which lacked both the cytochromes *c*_Z_ (PscC-1 and PscC-2) and the second FMO trimer subunits.

The core molecular architecture of the GSB complex is also homologous to the known Type I RC structures (Figure 3C) in which the PscA subunits form eleven transmembrane α-helices, with the first six binding the antenna BChls and the remaining five associated with the RC cofactors [4,5]. Along with the 82-kDa core PscA-1 and PsaA-2 subunits and the two accompanying, asymmetrically positioned FMO trimers, four additional subunits were assigned [4]: two identical 23-kDa cytochromes *c_Z_* (PscC-1 and PscC-2); the 24-kDa PscB housing the F_A_ and F_B_ [4Fe-4S] clusters, along with the associated 17-kDa PscD subunit (the PscC subunits as well as the second FMO trimer were absent from the initial structure [5], reflecting loss during purification following Triton X-100 solubilization). The cofactor assignments included 78 (B)Chls (48 BChls *a* bound to the two FMO trimers and 26 BChls *a* and 4 Chls *a* bound to the RC-PS); five lipids (two monogalactosyl diglycerides and two phosphatidylglycerol (PG) molecules [4,5], as well an additional PG modeled adjacent to the A-1 site [4] not seen in initial structure [5]); four carotenoids (two OH-chlorobactene glucoside laurates in contact with both the cytoplasmic and periplasmic BChl layers and two chlorobactenes associated with the cytoplasmic layer [5]); and 2 Ca^2+^ ions and 3 [4Fe-4S] clusters (F_X_, F_A,_ and F_B_) [4,5].

The C_2_ symmetry axis, as assumed by the GSB RC cofactors, is shown in Figure 3D, together with the coordinating residues provided by the homodimeric (PscA)_2_ protein [5]. The essentially identical distances between the cofactors of each branch suggest that electron flow occurs on both sides. Note that the distances between A_0_ and F_X_ of 18.1 and 18.2 Å [4] (reported as ~17.5 and 17.6 Å in [5]), which are shorter than the ~23 Å observed in PSI [7] and similar to the 17.9 Å for the HB RC [3]. Thus, the A_0_ and F_X_ components of the GSB RCs are also sufficiently close to promote the observed slowed electron transfer rate [31]. While a PQ molecule intervenes in sustaining electron transfer between A_0_ and F_X_ in PSI, and an inserted MQ was shown to play such a role in the HB RC under high-light intensity conditions [21], only trace levels of Q were found in the GSB core RC-PS complex [5]. Moreover, it was not possible to model an MQ into an appropriate binding site in the GSB complex, and therefore, general agreement exists that electrons are transferred directly from A_0_ and F_X_ (see, however, [32,33,34]).

The residues serving as the coordinating ligands for the F_X_ cluster were identified as Cys527 of PscA-1 and Cys 536 of PscA-2, located at the interface between the two PscA subunits [5] as seen in other Type I RCs [3,6,7]. The PscB-associated F_A_ and F_B_ clusters are located at the cytoplasmic surface along with the PscD subunit (Figure 3A,B) in positions similar to the F_A_ and F_B_ containing PsaC and the associated PsaD in the PSI complex. While PscB has a similar structure and electron transfer role as PsaC, they show little sequence homology, as also observed between PsaD and PscD, suggesting that the PsaC and PsaD components of the heterodimeric PSI RC arose by convergent evolution. The associated PscB and PscD proteins also interact with the FMO and RC core and are in positions to contribute to the stability of the FMO-RC-PS complex [5].

A comparison of the Type I GSB RC-PS with the PSII complex (Figure 4A,B) shows that the majority of the 24 BChls *a* serving as antenna pigments exhibit binding sites that are largely conserved in each of the 12 Chls *a* associated with the Chl protein (CP)43 and CP47 PSII core antennae, among their respective 13 and 16 Chl *a* molecules [16]. Moreover, an essentially complete overlap occurs in the locations of the six GSB RC (B)Chl cofactors with those of the HB and PSI RCs (Figure 4C,D), together with good agreement in the positioning of the 24 common antenna (B)Chls, with only a few minor differences in locations. The additional (B)Chls found in the HB RC-PS and in PSI are mainly located at the peripheries and the gaps in each pigment cluster.

The similar arrangement of the (B)Chl in Type I and II RCs (Figure 4) extends to the RC core of the purple chlorophototrophic bacteria, together with the conservation of the structural architecture of their apoproteins [14,15]. This reflects common evolutionary histories, whereas the differences in the types of both additional core and peripheral antennae reveal distinct events in evolution. This is seen in the RC-PS for both the HB and the GSB, as well as in PSI and PSII, where such differences signify their distinct phylogenic status. Thus, early in the course of evolution, the lack of pigments associated with the GSB RC-PS was compensated for by the presence of the FMO and chlorosomes, acting as peripheral antennae. Moreover, additional pigments are present in the Type I RCs of HB and PSI, while for PSII, the core LH pigments are contained within the CP43 and CP47 proteins. While the LH1 antenna rings of purple bacteria house the core LH pigments, distinct peripheral LH complexes designated as LH2, LH3, and LH4 functions as accessory antennae in many anoxygenic phototrophs [35,36]. For the oxygenic phototrophs, the phyco-bilisomes in cyanobacteria and red algae and the LHCII and LHCI complexes of higher forms, such as land plants and green algae, also function in these accessory roles [1].

It is also noteworthy that when viewed parallel to the membrane plane, a similar layering of (B)Chl distributions was seen for the GSB RC-PS as that observed for the HB complex, as well as in PSI and PSII [3,4,5,6,7,16]. Since the pigment distances observed between the two layers of these antenna components are longer than those within each layer, energy transfer would be slightly less efficient in the former circumstance. This problem is overcome in the HB RC-PS and in PSI and PSII by (B)Chls that act as linkers between the two pigment layers having shorter Mg-to-Mg distances (9.7–11.7 Å), allowing more rapid interlayer energy transfer. Since the GSB complex lacks such linker BChls, the energy transfer rate between the two layers is consequently slowed. Nevertheless, a carotenoid derivative modeled as OH-chlorobactene glucoside laurate was shown to connect the two pigment layers [4,5] and is capable of transferring energy to the BChls while also functioning in their quenching but is unable to transfer energy between the BChls in the two layers [5].

The structure obtained for the GSB RC-PS complex allows for an assessment of the low-efficiency energy transfer pathway from the FMO BChls to the GSB antenna assembly [27,28,29,30]. BChl-3 molecules provided by the FMO-1 and FMO-2 are located in positions to pass excitons across 21.5 and 21.7 Å gaps to respective PS LH-BChls designated as BChls A808 and A810 of PscA-1 [4,5], located near the outer surface of the cytoplasmic pigment layer (Figure 5A). On the PscA-2 side, the BChl-3 provided by FMO-4 is positioned to pass excitons across a 27.0 Å gap onto BChl *a*807, while for the BChl-3 molecules provided by FMO-5, paths of 23.4 and 22.5 Å to BChl *a*807 and BChl *a*808, respectively, were seen [4] (Figure 5B). In contrast, while efficient energy transfer occurs between a gap of 8 to 13 Å between LHCI-Chls and PSI antenna pigments [37,38] as aided by numerous Chls occupying the gap space, in the case of the GSB RC-PS, no such intragap BChls were found, and the resulting much longer distances clearly account for the low efficiency of energy transfer from FMO to the core RC-PS. Since BChls A811 and A815 of the GSB PS are located closest to the RC P_840_ BChl special pair (edge-to-edge distances of 11.5 and 13.0 Å [4], Mg-to-Mg distances of 20.4 and 18.9 Å, respectively [4] (Figure 5C)), they are in a position to perform the final energy transfer step. The energy transfer distances provided by these structural considerations provide a blueprint for future theoretical studies aimed at further elucidating relevant pathways culminating in exciton transfer to P_840_, facilitating the initiation of the RC charge-separated state.

Duplicate copies of the unique cytochrome *c*_Z_ (PscC-1 and -2) are associated with the GSB RC (Figure 3A,C). They exist as integral membrane proteins with two TMHs and function in the re-reduction of the photooxidized P_840_^+^ BChl special pair using electrons obtained from the oxidation of reduced sulfur compounds [39]. Due to the flexibility of the heme *c*-containing C-terminal portion of PscC, which fluctuates during electron transfer between the cytochrome *bc*_1_ complex and P_840_ [40], this domain could not be modeled into the RC structure [4]. Instead, the crystal structure of the *Cba. tepidum* cytochrome *c*_Z_ electron carrier domain [41] was utilized in a protein–protein docking analysis, providing a detailed picture of the P_840_ BChl special pair re-reduction site (Figure 6A). Although both PscC-1 and PscC-2 are associated with the PscA homodimer, only a single heme *c* binding site was revealed [4], similar to the situation in the cyanobacterial PSI [42] and the RC of purple bacteria [43]. The parallel PscA-1 and PscA-2 Trp residues (Trp601) are associated at the docking interface with heme *c*, located in the PscC crevice, and are positioned at an edge-to-edge distance of 2.4 Å from the heme *c* propionate group (Figure 6A). Within the PscC docking site, the exposed PscC Lys168 and Arg181 residues at the heme edges interact with the negatively charged PscA surface. A similar set of twin Trp residues, separately provided by PsaA and PsaB in the cyanobacterium *Thermosynechococcus elongatus*, participates in the docking between PSI and cytochrome *c*_6_ [42]. In addition, the shortest observed center-center distance between the cytochrome *c*_Z_ heme and P_840_ of 21.9 Å (Figure 6A) compares favorably with the distances observed for the hemes of *T. elongatus* cytochrome *c*_6_ [42] and the *Cereibacter sphaeroides* (formerly *Rhodobacter sphaeroide*s) cytochrome *c*_2_ [43] and their primary RC donors (21.1 Å and 21.3 Å, respectively). Despite these advances in elucidation of the mechanism of GSB RC special-pair reduction by cytochrome *c*_Z_, the presence of parallel pathways of energy transfer to P_840_, while only a single site exists for the re-reduction of this donor site, leaves open the question of how the two cytochromes communicate in supplying necessary electrons [4].

As first observed for the ferredoxin binding site of PSI, which includes basic residues provided by both PsaA and PsaC [44], participating residues provided by the GSB-RC (Figure 6B) include Lys149 and Lys153 of PscB, as well Lys45, Arg47, and Arg48 of PscA-1; each of these residues are grouped in the vicinity of the F_B_ [4Fe-4S] cluster [4]. While an insertional mutant of the *Cba. tepidum psc*D gene showed a much slower rate of energy transfer to P_840_ from the antenna pigments, the photoreduction of NADP^+^ by ferredoxin was only slightly diminished [45], apparently reflecting an additive relationship of PscD to the PscB catalyzed ferredoxin reduction. Nevertheless, interactions between PscB and PscD are thought to promote the docking of ferredoxin while also contributing to the electron exit site [4].

Overall, these crucial structural studies have revealed the surprising finding that the positioning of the LH-BChls in the Type I GSB RC-PS homodimer most closely resembles that of the PSII antenna, in which the core GSB LH-Chls occupy two separate clusters similar to that provided in PSII by the CP43 and CP47 proteins located on either side of the electron transferring cofactors associated with the respective D1 and D2 proteins. Notably, both LH domains provided by PscA of the GSB-RC and the PSII core LH proteins contain fewer core antenna pigments than the HB RC-PS and PSI. Importantly, the recent recognition that the PSII H_2_O oxidizing Mn_4_CaO_5_ cluster is located in a homologous position to the Ca^2+^-binding site associated with the bacterial Type I RCs, which also possess a number of structural similarities, further supports a close relationship between these homodimeric structures and PSII [2]. These key findings [4,5] provide additional examples of how the elucidation of new structural details of the Type I homodimeric RC-PS complexes contributes to our understanding of the origins of heterodimeric PS. Moreover, the detailed molecular architecture of the entire GSB RC-PS [4] provides an incentive for further experimental and theoretical examination of the energy and electron transfer steps catalyzed by this photosynthetic system.

### 2.3. Structure of the Reaction Center-Photosystem of the Chloracidobacteria

*Cac. thermophilum* was first recognized in a metagenomic analysis of microbial mats retrieved from an alkaline hot spring, representing the earliest identified phototrophic member of the phylum Acidobacteria [46]. The Type I homodimeric RC-PS isolated from *Cac*. *thermophilum* [47,48] has a unique pigment content consisting of three forms of (B)Chl, including Zn-BChl a’ [49]. FMO and chlorosomes are also present as peripheral antennae [50], as seen for the GSB [26,27,28].

The structure of the Type I RC-PS of *Cac. thermophilum*, recently determined at 2.6 Å resolution by cryo-EM [6], is shown in Figure 7A,B. Besides the homodimeric PscA-1 and PscA-2 subunits, eight additional subunits were identified, which included PscB, housing the terminal [4Fe-4S] cluster electron acceptors F_A_ and F_B_, accompanied by PscZ, with an apparent PscB stabilizing role (Table 1). Also included were the distinct membrane-associated monoheme cytochrome *c* subunits PscX and PscY, forming a heterodimer in which their respective hemes *c* 1 and 2 are located within deep protein grooves where they serve as the secondary and primary electron donors to the RC P_840_ special pair. In addition, low mol. wt. subunits Psc U, PscV, PscW, and UPP of unknown function were also identified. However, the FMO antenna was lost during purification, and its structure was not determined along with the *Cac. thermophilum* RC-PS.

The RC pigments and cofactors consist of the unique P_840_ (Zn-BChl *a*’ special pair), A_CC_ (Chl *a*), A_0_ (Chl *a*), and the terminal [4Fe-4S] electron acceptors (F_X_, F_A_, and F_B_) (Figure 7C). NMR studies [51] have established Chl *a* as the primary RC electron acceptor, while HYSCORE spectroscopy, together with DFT analysis [52], has confirmed the role of the Zn-BChl *a*’ special pair as the primary RC donor. Two phosphatidyl-N-methylethanolamine (PME), and notably, 2 Ca^2+^ ions were also associated with the RC [6], and possible roles of the requisite prokaryotic Ca^2+^-Type I RC binding sites in the evolution of the PSII H_2_O oxidizing Mn_4_CaO_5_ cluster are discussed in detail below. Additional molecules identified in other regions of the structure included 16 BChls *a*, 2 Chl *a*, 10 lipids (4 PMEs, 4 diacylglycerylhydroxymethyl-N,N,N-trimethylalanines, plus 2 phosphatidylethanolamines), 2 lycopenes, and 6 H_2_O molecules.

The LH-(B)Chl *a* content consisting of 6 LH-Chls *a* and 16 LH-BChls *a* provides the *Cab. themophilum* RC-PS with the ability to harvest both visible and far-red light, giving rise to an excitonically coupled “hybrid” LH system [6]. The boxes in Figure 8A demarcate pigment distribution in the cytoplasmic and periplasmic bilayer leaflets, with the PscA-2 cytoplasmic leaflet harboring one Chl *a (*913) and five Bchls *a* (902, 903, 904, 905, and 907) (Figure 8B), while two Chls *a* (912 and 914) together with three BChls *a* (906, 908, and 909) are found in the corresponding periplasmic leaflet (Figure 8C). The Mg^2+^-to-Mg^2+^ distance of 15.0 Å between BChl *a*902 and BChl *a*909 represents the closest inter-bilayer connection and serves as a common energy transfer locus for the distinct hybrid pigment arrays (Figure 8A–D) [6]. Based upon the Mg^2+^-to-Mg^2+^ distances between BChl *a*902 and the cytoplasmic array of BChls *a* ranging from 12.8 to 17.9 Å and 9.6 Å to Chl *a*913 (Figure 8B), a pathway of energy transfer was posited between these pigments that culminate at BChl *a*909 within the periplasmic layer (Figure 8D). Once the energy reaches BChl *a*908, the transfer occurs directly to P_840_, representing the PscA-1 half of the Zn-BChl *a*’ special pair, but it is also expected to occur simultaneously from Chl *a*914 (Figure 8C), thereby completing this efficient excitation energy scheme within the bilayer leaflets.

In addition to individual lycopene molecules associated with the LH-BChls *a* in each PscA subunit (Figure 8A), a 22-kDa carotenoid-binding protein is also present in *Cac. thermophilum* [46]. Although absent from the RC-PS structure [6], the carotenoid-binding protein may serve as both a peripheral antenna, passing visible excitations onto LH-BChls, and in a photoprotective role exerted against reactive oxygen species, that maintains survival when exposed to high O_2_ levels during the diel cycle in the microbial mat environment.

The membrane-anchored PscX and PscY subunits of the CB RC, which function in the serial donation of electrons to photooxidized P_840_, are positioned with edge–edge heme distances of 2.3 Å, while the PscY heme 2 is located at a distance of 10.4 Å from the Zn-BChl *a* P_840_ special pair [6]. Distinct from the GSB-Type I RC, in which the immediate RC-donor *c*-type cytochrome (cytochrome *c*_Z_) consists of a homodimer of the integral-membrane PscC-1 and PscC-2 subunits, with each containing two RC-associated TMHs (Figure 3C), PscX and PscY form a heterodimer associated with the RC-PS via both intramembrane regions and the periplasmic membrane surface. Associations within the membrane occur via hydrophobic interactions involving loop domains of PscX (residues 16–25) and TMH1 of PscA-1 and via PscY (residues 15–24) entailing both TMH1 and TMH11 of Psc-A2. In addition, N-terminal Cys residues (PscX-Cys21 and PscY-Cys20) are subjected to post-translational modifications that form linkages modeled as hydrocarbon chains (designated as Chain_Cys21_ and Chain_Cys20_) that assist in anchoring the cytochromes to the RC. At the peripheral membrane region, PscX and PscY are associated with PscA-1 and PscA-2 via numerous H-bonds linkages as well as salt bridges in which the Asp55 and Lys346 of each PscA subunit play crucial roles in spite of the asymmetry of the PscX-PscY heterodimer (Table 1) [6].

In contrast to the above associations existing between monoheme cytochromes *c* and homodimeric Type I RCs, the RCs in many purple bacteria are directly reduced by tetraheme cytochromes *c* attached to the membrane by means of lipid anchors covalently linked to N-terminal Cys residues [9,10,11], which arise after the removal of N-terminal signal sequences by the post-translational truncation process. These lipid anchors were first identified as a thioester-linked diacylglycerol molecule in *Blastochloris viridis* [53,54], and subsequently, a triacyl structure was found in *Thermochromatium tepidum* [55] consisting of an *N*-acyl chain and an *S*-diacylglycerol linked to the N-terminal Cys. However, the purple chlorophototrophic extremophile *Rhodopila globiformis*, capable of growth at pH ≥ 3.0, lacks this Cys within the N-terminal region. Consequently, the unprocessed, full-length tetraheme cytochrome *c* is attached to the membrane via a membrane-spanning N-terminal ΤΜH [56]. Interestingly, this membrane anchoring helix shows significant structural similarity with the N-terminal domain of the RC-associated PufX polypeptide, responsible for establishing the LH1 ring gap, facilitating ubiquinone exchange between the RC and the cytochrome *bc*_1_ complex of *Ceb. sphaeroides* [57] and in *Rhodobacter* and related species [9,10,11].

Comparative sequence analyses of the N-terminal regions of the *Rpi. globiformis pufC* gene (encoding the tetraheme *c*-type cytochrome) and the *Ceb. sphaeroides pufX* gene, which share a common *puf* operon location immediately downstream from *pufM* encoding the RC-M subunit, demonstrated a phylogenetic relationship with the PufX protein as well as with *pufC* genes of related species forming tetraheme cytochromes in which N-terminal signal peptides are replaced with a lipid N-terminal membrane anchor [58]. Accordingly, a full-length tetraheme cytochrome via partial deletions within an ancestral *pufC* gene may represent the original archetypes for both the truncated cytochrome forms and the PufX protein [58]. It should also be noted that in the filamentous anoxygenic phototroph *Roseiflexus castenholzii* [59], attachment of the tetraheme *c*-type cytochrome component is also via a single N-terminal TMH anchor, albeit with additional associations at the periplasmic surface with both RC and LH proteins [60,61]. Importantly, *Rfl. castenholzii*, a green-nonsulfur bacterium lacking chlorosomes, as a member of the *Bacteroidata* (formerly *Chlorobiota*), is among the earliest anoxygenic phototrophs to emerge, based upon its position in a deep eubacterial branch of the phylogenetic tree of 16S rRNA gene sequences [62].

Among the numerous genes encoding *c*-type cytochromes in the *Cac. thermophilum* genome [63], Cabther_A2183 and Cabther_A2184 were assigned to the respective PscX and PscY cytochrome *c* subunits, which correspond to their 189 and 221 amino acid (AA) gene products (GenBank ID. AEP129221.1 and AEP12922.1), exhibiting a ~30%. AA SI. This dicistronic operon is situated immediately downstream from the *pscAB-fmo* operon and was expected to be the source of the RC-associated cytochromes *c* [63], as ultimately confirmed in the RC-PS structure [6]. In addition, a second dicistronic operon (Cabther_A1370 and Cabther_A1371 was found whose respective gene products AEP12121.1, (158 AA, ~34% SI to PscY) and AEP1212.1 (119 AA, ~42% SI to PscY,) also form putative membrane-associated proteins. Other located genes include Cabther_A0280, which encodes a soluble 117 AA protein (AEP11047.1, ~30% SI to PscY), along with a protein annotated as the 190 AA subunit III of cytochrome *c* oxidase cbb3 ABV27349.1, 84% SI to PscX) in which a *c*-type heme CXXCH binding motif was found.

Further BLAST searches of the *Cac. thermophilum* Type B laboratory isolate genome [63] revealed an apparent *c*-type cytochrome ortholog with PDB ID: 7VZR_c (145 AA, 100% SI with PscX, while lacking the 15 N-terminal and the 29 C-terminal residues); the origin of this puzzling sequence redundancy requires further investigation. Other identified cytochromes *c* included the putative monoheme *c*-type cytochrome subunit of a quinol-cytochrome *c* oxidase (AEP13240.1, 213 AA, ~32% SI to PscY). Also, a *petAB* operon is present in which a downstream gene (Cabther_A 0912) encodes a multifunctional cytochrome *cyd*-type quinol oxidase protein (AEP11669.1, 483 AA) containing a C-terminal *c-*type cytochrome region, apparently fulfilling the role of cytochrome c_1_ which is missing from the *petAB* operons. This cytochrome *c* region is attached to the N-terminal membrane *cyd*-integral domain by a 70-residue linker. Moreover, a second *petAB* operon is present that contains a *petB* gene whose multifunctional product contains a C-terminal cytochrome *c* domain with a similar unusual membrane linkage.

The possibility has been proposed that these multiple oxidoreductases, together with an Alternate Complex III, function in accommodating the wide-ranging environmental conditions existing during the diel cycle in the microbial mat community [63]. These include extensive changes in O_2_ availability ranging from O_2_ supersaturation under intense sunlight to the anoxic conditions that prevail during the late afternoon and evenings. Clearly, as a homodimeric Type I-RC containing aerobic anoxygenic phototroph, *Cac. thermophilum* provides a unique system for obtaining a more detailed picture of the evolution of the oxygenic phototrophs.

## 3. Further Insights into the Evolutionary Origins of Oxygenic Photosynthesis Pro-Vided by the Structures of the Homodimeric Type I Reaction Center-Photosystems

A variety of structural and functional features of the Type I-homodimeric RC-PSs of the HB, GSB, and CB have afforded valuable insights into the evolutionary origins of both the PSI and PSII heterodimeric RCs of higher phototrophs. The major relevant discoveries, as discussed recently by Gisriel et al. [64], include: (i) structural similarities between the conserved Ca^2+^-binding sites found in the homodimeric Type I RC structures and the Mn_4_CaO_5_ cluster as well as in Tyr_Z_ donor site of PS II; (ii) the presence of a PG molecule involved in the ligation of the A_0_ cofactor of the GSB electron transfer chain, associated with conserved Arg residues in both the HB and GSB RCs which has possible implications for the development of Q exchange and reduction mechanisms; (iii) the similarities in the positioning of antenna (B)Chls within PS II and the GSB RC-PS and SI within these regions [2]; and (iv) the presence of a GSB excitation energy quenching mechanism promoted by a glycosylated carotenoid which likely forms the basis for the O_2_ tolerance processes that served to support the evolutionary origins of oxygenic photosynthesis. These findings hold promise for directing further investigations into both the evolutionary diversity of RCs and the relationships that led to the emergence of H_2_O oxidation in oxygenic phototrophs, largely responsible for the origin of an O_2_-containing atmosphere that facilitated the subsequent evolution of further life on earth.

Structural similarities between the sites on the homodimeric HB RC and the PSII Mn_4_CaO_5_ cluster, together with the respective Tyr_Z_ and Tyr_D_ donor sites of the D1 and D2 polypeptides of PSII, are shown in Figure 9A,B. It is seen that the recently recognized Ca^2+^ binding sites of (PshA)_2_ [3,64] and (PscA)_2_ of the GSB [4,5] and CB [6] RCs overlap (Figure 9C) and are located in structurally homologous positions to the Mn_4_CaO_5_ cluster of the D1 RC polypeptide. In addition, the Tyr_Z_ donor site of D1 (Tyr161) is H-bonded to His190, which has a counterpart in Tyr_D_ of the D2 polypeptide, where an H-bond exists between Tyr160 and His189 (Figure 9A), considered as evidence for a homodimeric origin of the two PSII regions [2]. This scenario holds that H_2_O oxidation was initiated in two symmetrically placed catalytic cluster sites on each side of the RC (see below) as supported by ligands found in both the D1 and D2 polypeptides and their respective CP43 and CP47 antennae, and ultimately culminating in the assembly of the active Mn_4_CaO_5_ cluster on the D1 side.

Parallels are shown in Figure 9D between the position of the PshA Ca^2+^-binding site and the D1 Mn_4_CaO_5_ cluster in which Asp468 of PshA-1 is seen coordinated to the Ca^2+^ atom located in a region comparable to the D1 Tyr-His redox pair and adjacent to the D1 Mn_4_CaO_5_ cluster. Similarities in locations are also observed between the C-termini of the PshA-1 and D1 polypeptides, where the PshA-1 C-terminal Val608 residue is liganded to the Ca^2+^ atom (Figure 9E,F), while for D1, the Ca^2+^ atom within the Mn_4_CaO_5_ cluster is liganded to the C-terminal Ala344 (Figure 9G). Furthermore, Asn263, positioned within the peripheral domain of the PshA-1 antenna region between the 5th and 6th TMH (Figure 9F), is associated with the Ca^2+^-binding site via two H_2_O molecules. Likewise, Glu354 is also located between the 5th and 6th TMH of the CP43 peripheral domain and is linked to the D1 Mn_4_CaO_5_ cluster (Figure 9G). It is also noteworthy that PSI lacks a Ca^2+^ binding site (Figure 9E), as the C-terminal region of helix 11 of PsaB is two turns greater in length than in the PshA counterpart. While these structural comparisons have contributed to our understanding of the origins of the H_2_O oxidation process (see below), they do not explain the source of the Pheo-Q Type II electron transfer chain of PSII, which has instead been suggested to have occurred after the heterodimerization process [12], as previously discussed in Section 2.1. It is therefore likely that PSII arose from a chimera of an H_2_O-oxidizing complex and the RC-electron transfer chain [65]. Such an ancestor would have served in the origin of extant RCs that evolved into Type I RCs of the oxygenic phototrophs by incorporating more pigments and into Type II RCs by losing the Fe-S cluster, which was replaced by Q as the terminal electron acceptor.

Cardona and Rutherford [2] additionally posited that the increased redox potentials (“oxidative jumps”) attained on the donor and acceptor sides leading to functional PSII electron transfer and H_2_O oxidation were achieved by means of crucial mutations that occurred in a common homodimeric ancestor resulting in the pronounced structural changes responsible for simultaneous oxidative jumps. Following these large structural adaptations, a Tyr residue adjacent to the Ca^2+^ binding site located in the position near the D1 Y_Z_ donor (Figure 9F,G) was capable of generating a tyrosyl radical with the ability to oxidize aqueous Mn^2+^ cations. The bound Ca^2+^, together with the C-terminal carboxyl group and a ligand formed with the antenna domain, already existing in the HB RC-PS structure (Figure 9F), stabilized the oxidized Mn. Subsequently, protons released from the bound H_2_O molecules by the Mn oxidation process promoted μ-oxo-bridge formation between adjoining Mn^2+^ cations as reflected in the assembly factor-free photoactivation process realized in the case of the PSII Mn_4_CaO_5_ cluster [64]. Moreover, the ability to oxidize H_2_O is thought to have occurred prior to the emergence of both PSI and the Type I homodimeric RCs, which required the presence of a system for the avoidance of reactive oxygen species [2]. These considerations, as well as a general evolutionary scheme for RCs, are shown in Figure 10.

It is important to note that for the GSB, each PscA subunit contains 12 LH-BChls situated in positions conserved in the HB Type I PS, as also seen for the LH-Chls found in both PSI and PSII (Figure 4). This is consistent with the possibility that these GSB antenna BChls occupy sites found in the ancestral RC-PS complex. The HB complex possesses an additional 13 LH-BChl *g* molecules associated with both PshA subunits, compensating for the lack of peripheral antennae, as exhibited by the presence of chlorosomes and FMO in the GSB. Two of the additional BChls *g* in the HB complex are located closest to the electron transfer chain (Figure 4C); the absence of BChls in these sites accounts in part for the poor coupling of the bulk GSB antenna to the RC. Notably, the shared features of the antennae of the GSB core PS and PSII [64] imply an early origin for the H_2_O oxidation process. Moreover, the fact that the GSB RC-PS possesses features found in both Types I and II RCs likely reflects an origin for the GSB complex from a common homodimeric RC-PS ancestral complex.

The GSB RC-PS associated chlorobactene carotenoid designated as F26, bound in a position adjacent to the core complex interface with FMO [5], was shown to be involved in both energy quenching and transfer processes [66]. Accordingly, a slowing of fluorescence quenching of the core LH-BChls was demonstrated in a *Cba. tepidum* mutant unable to glycosylate carotenoids, despite the lack of apparent effects on photosynthetic apparatus structure. In addition to inhabiting anaerobic environments, *Cba. tepidum* is microaerophilic, supported by the ability of the glycosylated carotenoid components to facilitate tolerance to low O_2_ levels. An additional glycosylated carotenoid glycoside, F39 (hydroxy-γ-chlorobactene glucoside laurate), specific to the core complex, is located in a region that is structurally conserved in PSII and binds both carotenoids and lipids, while also participating in the quenching of Chl excitation energy. Importantly, since the quenching of excitation energy by F39 has provided a basis for O_2_-tolerance in GSB, the structural conservation found between regions inhabiting GSB F39 and PSII has suggested that the development of such energy quenching process served as a potent driving force in the very early stages of RC evolution [64].

Overall, these recent evolutionary analyses point to the origins of an O_2_ evolving complex and a functional PSII very early in the history of the evolution of life on earth and further suggest that both RC types arose from a primordial ancestor that already contained Types I and II RCs [2]. This is illustrated in Figure 10, where the proposed stages in the evolutionary history of extant Type I and Type II RCs are shown in terms of a step-by-step depictions from the most recent common ancestral precursor of both types of RCs via a series of separate subsequent homodimeric evolutionary precursors, ultimately leading to the heterodimerization process. It is likely that cyanobacteria diverged early and prior to extant phyla, which include anoxygenic phototrophs [67,68]. Consequently, anoxygenic Type II RCs appear to have evolved independently subsequent to their oxygenic counterparts, with the Type I/Type II split occurring prior to the diversification of anoxygenic phototrophic bacteria [69]. This runs counter to the earlier but largely unsupported views that anoxygenic Type II RCs are more ancient than the Type II RCs of their oxygenic counterparts [70] and that the anoxygenic to oxygenic photosynthetic transition occurred when a PSI containing cyanobacterium formed PSII with the ability to oxidize H_2_O (summarized by Olson and Blankenship [71]). Instead, the early origins of both PSI and II are now supported by a considerable body of evidence [2,12,13,64,65,66,67,68,69,72,73] forming the basis for the proposal that the gene duplication event, resulting in the heterodimerization of a homodimeric ancestor of PSII into distinct D1 and D2 polypeptides likely predated the subsequent evolution of the well-described L and M polypeptides of anoxygenic Type II RC [69,71].

Recently, a homodimeric Type I RC was unexpectedly discovered in the newly described filamentous bacterium *Canditatus* Chlorohelix allophototropha [74], as revealed in a metagenomic analysis of a cultured anoxic region of a stratified H_2_O column arising from a boreal forest lake. As a member of the phylum *Chloroflexota* in which only Type II RCs had previously been described [75,76], the finding of Type I RCs in Ca. Chx. allophototropha significantly alters prevailing views of RC diversity within the eubacteria. A detailed phylogenomic analysis [74] has suggested that the most recent common phototrophic ancestor of both RC classes in the *Chloroflexota* contained genes encoding BChl synthesis and at least one RC type. Thus, despite the differences in RC types in Ca. Chloroheliales and the remaining members of this phylum, likely both share a common phototrophic genetic heritage.

Two models were advanced to account for the manner in which the phototrophic ancestor gave rise to both Type I and Type II RCs in the *Chloroflexota* [74]. (i) The genetic displacement model [77] in which the progenitor contained either Type I or Type II RCs, while lateral gene transfer was responsible for the presence of the final RC type. The genes for BChl synthesis of the common ancestor were retained, and the new RC then became operational, with its genes displacing the original RC genes. Since chlorosomes are only found with Type I RCs, apart from the *Chloroflexota*, but also exist in Ca. Chloroheliales, which is located basal to the *Chloroflexota* Type II RC-containing clade, the phototrophic ancestor would likely have contained Type I RCs in this scenario. (ii) The differential genetic loss model in which the most recent common ancestor carried both Type I and Type II RCs with only the Type I RCs retained in Ca. Chloroheliales, while Type II RCs were retained in the remaining *Chloroflexota.* While such a precursor containing the two RC types is also thought to have driven the evolution of the oxygenic phototrophs, they were combined into a single electron transfer pathway, ultimately forming the Z-scheme in which PSII feeds electrons into PSI, rather than being subjected to the Type I or Type II RC loss as suggested for the evolution of anoxygenic phototrophs such as in Ca. Chloroheliales and other *Chloroflexota* orders. Moreover, despite the parallel evolutionary histories proposed for the *Chloroflexota* and *Cyanobacteria* (Figure 10), the latter remains the sole bacterial phylum containing oxygenic phototrophs. 

The high-resolution structures of the Type I RC-PS homodimers of *Hmi. modesticaldum* [3], *Cba. tepidum* [4,5] and *Cac. thermophilum* [6], represent three bacterial phyla (the *Firmicutes, Bacteroidata* (formerly *Chlorobiota*), and *Acidobacteriota*, respectively. Additional phyla containing anoxygenic phototrophs include the Proteobacteria and Gemmatimonadati, where a number of high-resolution structures are available for their Type II RCs as surrounded by a variety of open or closed elliptical LH proteins [9,10,11], as well as the Chloroflexota possessing either Type II RCs or the newly discovered homodimeric Type I RC [74]. Also included is the phylum nov. Vulcanimicrbiota [78], in which a metagenomic-assembled genome of a bacterium initially designated as *Ca.* Eremiobacterota (WPS 2) [79,80] has demonstrated the presence of a Type II RC lacking the H subunit as in the Chloroflexota, but also including a novel sixth TMH in the RC-M subunit, as well as substantial differences in residues acting as ligands to the RC cofactors [79]. These findings extend a novel and unusual Type II RC to this new phylum, which also has importance in the ultimate understanding of the origins of anoxygenic phototrophs.

## 4. Conclusions

The high-resolution homodimeric Type I RC-PS structures described here for three distinct species of anoxygenic phototrophs [3,4,5,6] have provided a greatly improved understanding of both the basic mechanisms driving the earliest types of phototrophy and the evolutionary events leading to the extant Type I and Type II RCs. Each of these resolved Type I RC-PSs exhibit the two-fold symmetric distribution of electron transferring cofactor as well as the basic 11-ΤΜH structure of PSI [7], in which the five C-terminal TMHs house the RC domain and the six N-terminal counterparts contain the LH pigments [3,4,5,6]. While all three structures lacked Q molecules, the distances between their A_0_ Chl *a* molecules and F_X_ [4Fe-4S] clusters of 17.9–18.2 Å [3,4,5,6], as compared to the 22.8 Å in PSI in which PQ intervenes as an intermediate electron acceptor [7], are sufficient to allow electron transfer to proceed between A_0_ and F_X_ at the observed intrinsic slowed rates.

The HB RC-PS is the simplest light–energy transducing complex yet described and contains two copies of only one additional subunit designated as PshX that serves as a low-energy antenna [19]. The GSB RC-PS structure contained four subunits in addition to the core Psc-A1 and Psc-A2 polypeptides and two FMO trimers [4]. These include the PscB terminal electron acceptor containing the F_A_ and F _B_ [4Fe-4S] clusters, the associated PscD subunit, and the two cytochrome *c_Z_* subunits PscC-1 and PscC-2. Although the FMO antenna was lost during purification, the CB RC-PS structure proved to be the most complex with the core Psc-A1, Psc-A2 homodimer accompanied by eight additional subunits consisting of the F_A_ and F_B_ containing PscB subunit and its associated PscZ putative stabilizing-protein, the cytochrome *c* electron donating PscX-PscY heterodimer, and four low mol. wt. subunits (PscU, PscW, PscV, and UPP) of unknown function. The increasing number of subunits found in these three homodimeric RC-PS structures (CB > GBS > HB) is consistent with their positions on the maximum likelihood phylogeny tree of type I RCs, as reported by Gisriel et al. [64].

A total of 54 LH-BChl *g* molecules were revealed in the HB RC-PS [3] as compared with the 26 associated LH-BChls *a* in the GSB RC-PS structure [4,5] and the 16 LH-BChls *a* and 6 LH-Chls *a* located in the CB RC-PS [6]. The extra HB LH-BChls *g* partially compensate for the lack of the FMO and chlorosome antennae as found in the GSB and CB. Moreover, the positioning of the LH-BChls *a* in the GSB RC-PS core exhibited a high degree of homology with the placement of the LHChls *a* in the CP43 and CP47 PSII antenna proteins, as also seen for the two LH-BChls located in the RC-associated positions of the PSII Chl_Z-D1_ and Chl_Z-D2_ sites [5]. The core GSB arrangement of the 24 GSB LH-BChls is also largely retained among the 54 HB LH-BChls *g* and the 79 PSI LH-Chls *a*. Importantly, this conservation of LH-(B)Chl arrangements observed between the GSB and PSII, together with the homologous localizations of the symmetrically positioned Type I-RC Ca^2+^ binding sites and the PSII Mn_4_CaO_5_ cluster as well as the Tyr_Z_ RC donor sites, underscores the close relationship found between homodimeric Type I and Type II PSII RC structures [64].

These similarities infer an early origin for the PSII Mn_4_CaO_5_ cluster catalyzing the O_2_ evolution process and also that the GSB RC-PS possesses attributes conserved in both Types I and II RCs, which implies that the GSB complex originated from a common homodimeric RC-PS ancestor [69]. Furthermore, the location of GSB glycosylated carotenoids in homologous positions to carotenoids found in PSII likely reflects a photoprotective O_2_-tolerance mechanism for supporting the initial stages in the evolution of oxygenic photosynthesis [64]. Duplication of the ancestral gene encoding a homodimeric Type I RC, together with genetic divergence, served to facilitate the emergence of the extant heterodimeric Type I and Type II RCs. This includes addition of a subunit housing the F_A_ and F_B_ [4Fe-4S] clusters in the structure of PSI [7] and a Q_A_ site on the D1 subunit and Q_B_ site on the D2 subunit in the emergence of PSII [12]. While the basic structure of PSII has remained essentially unchanged since the Archean era, the evolution of distinct PSII subtypes is a continuing process by virtue of genetic alteration in the D1 polypeptide that accommodate the variety of environmental changes to which higher phototrophs are exposed [81]. Finally, the unexpected finding of a homodimeric Type I RC-containing species of *Chloroflexota* [74], in which only Type II RCs had previously been described, represents a unique evolutionary transition that substantially alters current views of the diversity of photosynthetic organisms.

## Figures and Tables

**Figure 1 biomolecules-14-00311-f001:**
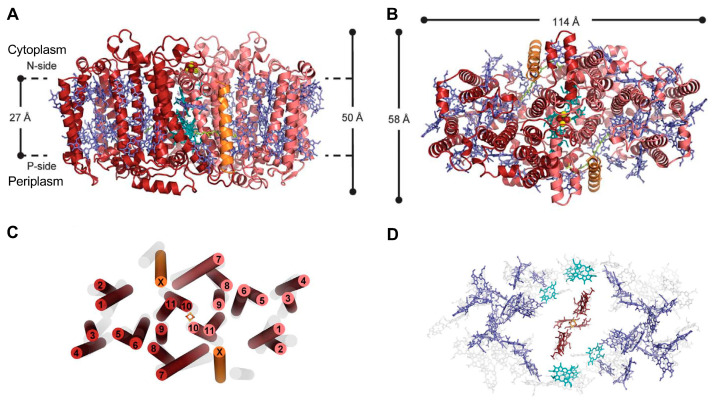
Structure of the Type I homodimeric reaction center-photosystem (RC-PS) complex of *Hmi. modesticaldum* as determined by X-ray crystallography at 2.2 Å resolution [3] (Protein Data Bank (PDB) ID: 5V8K). View parallel (**A**) and perpendicular (**B**) to membrane plane, later showing the cytoplasmic surface: PshA-1, red; PshA-2, pink; PshX, orange; cofactors (electron transfer), teal; antenna, blue; carotenoids, lime; [4Fe-4S], red and yellow spheres. Bacteriochlorophyll (BChl) and chlorophyll (Chl) tails are not shown. (**C**,**D**) Respective cytoplasmic views of transmembrane α-helices (TMH) and cofactor arrangements. PshA helices are numbered 1–11, with transparent gray helices corresponding to heterodimeric PSI helical arrangement and with [4Fe-4S] F_X_ component at the center. The PS pigments in panel D are superimposed and shown upon the gray PsaA-PsaB heterodimeric core-associated PSI cofactors. Heliobacterial (HB) electron transfer BChls and Chls, brown; bulk antenna pigments, blue; BChl *g* molecules flanking electron transport chain, teal.

**Figure 2 biomolecules-14-00311-f002:**
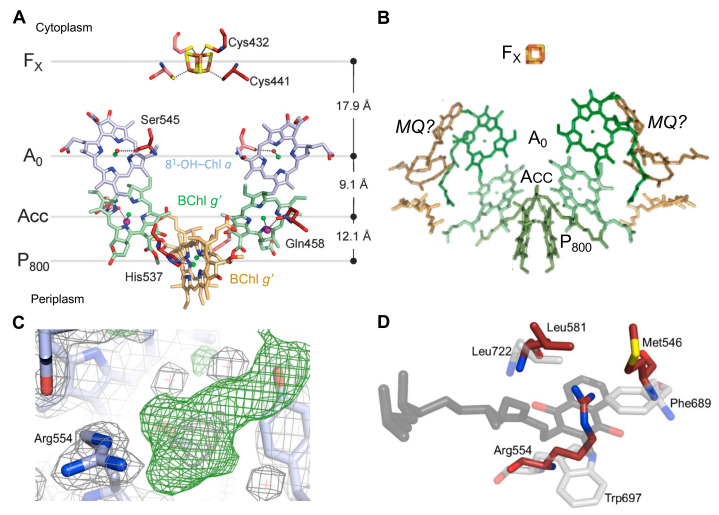
Cofactor arrangement of the HB RC electron transfer chain and modeled menaquinone (MQ) molecule inserted into a potential quinone (Q) binding site [3]. (**A**) Cofactors showing coordinating residues provided by the homodimeric (PshA)_2_. H_2_O molecules (small red balls) serve as axial ligands of the A_0_ primary acceptor, while the purple spheres liganded to the A_CC_ accessory BChls are possible chloride ions, and the magnesium atoms at the centers of the (B)chlorin rings are represented green spheres. Note that no carotenoid was found within the RC, and only two are seen in the (PshA)_2_ complex, a reflection of the anaerobic environment in which *Hmi. modesticaldum* is found, lacking reactive oxygen species. (**B**) Placement of MQ molecules in potential binding sites located with isoprenyl tails in positions similar to unassigned electron densities in the crystal structure [21]. (**C**) Unassigned electron density map in the vicinity of A_0_ of the HB RC [3] (Supplementary Material). (**D**) Residues (light gray carbons) provided by PsaA coordinate the phylloquinone (PQ) molecule of PSI (dark gray carbons) together with the residues of PshA (dark red carbons), forming an analogous binding site for MQ [3] (Supplementary Material).

**Figure 3 biomolecules-14-00311-f003:**
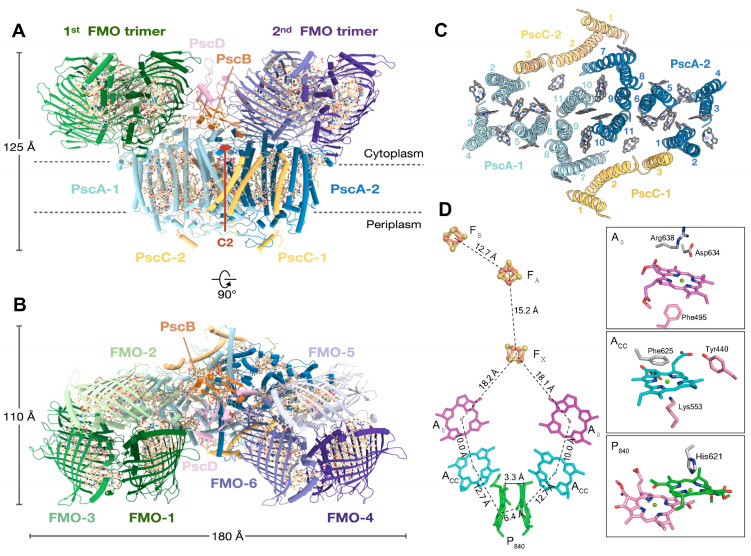
Overall asymmetric structure of the complete Type I homodimeric RC-PS complex of *Cba. tepidum* with associated trimeric Fenna-Matthews-Olson protein (FMO), as determined by cryo-electron microscopy (cryo-EM) at 2.5 Å resolution [4] (PDB ID: 7Z6Q). Views showing C_2_ symmetry of PscA components parallel (**A**) and perpendicular (**B**) to membrane plane also revealing the asymmetric arrangement of FMO components at cytoplasmic surface. The (B)Chls, carotenoids, and [4Fe-4S] cluster cofactors and lipids are all depicted in wheat. The distinct FMO-1 to FMO-3 and FMO-4-to FMO-6 components forming the respective FMO trimers are shown. (**C**) Periplasmic surface view of the arrangement of PscA and PscC helices, together with the RC-PS pigments. Helices are numbered starting from the N-terminal position: (B)Chls, in stick representation, gray; central Mg atoms, pink. (**D**) Cofactor arrangement of the *Cba. tepidum* RC electron transfer chain [3]. Residues coordinating the (B)Chl cofactors are shown at right: A_0_, Chl *a*; A_CC_, Chl *a*; and P_840_ special pair, BChl *a*’; PscA-1 residues, gray; PscA-2 residues, pink.

**Figure 4 biomolecules-14-00311-f004:**
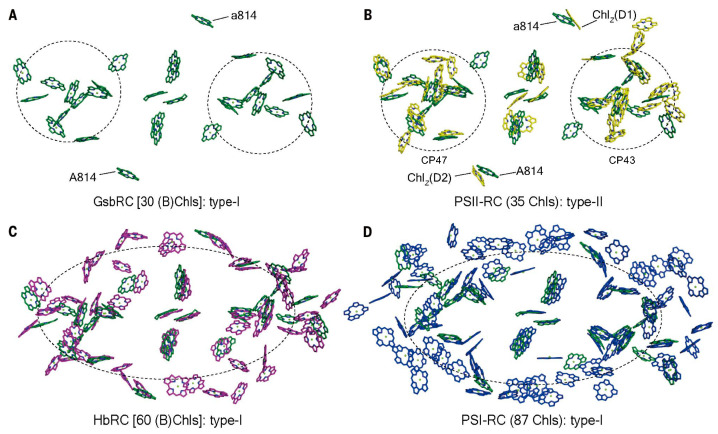
The common arrangement of (B)Chls in Type I and II RC complexes. (**A**,**B**) Distributions and superpositioning of (B)Chls in the green sulfur bacterial (GSB) RC-PS complex (dark green) and in PSII (yellow) shown perpendicular to the membrane plane [5]. Dashed circles demarcate similar clustering of the antenna (B)Chls around the RC electron transferring cofactors. (**C**,**D**) Distributions and superpositioning of (B)Chls in GSB RC-PS (dark green) with those of the HB RC-PS (purple) and PSI (blue). Dashed ellipses demarcate the closed rings of (B)Chls distributed across the two RC-PS subunits. Note that the (B)Chl distribution in the GSB RC-PS more closely resembles that of PSII than PSI.

**Figure 5 biomolecules-14-00311-f005:**
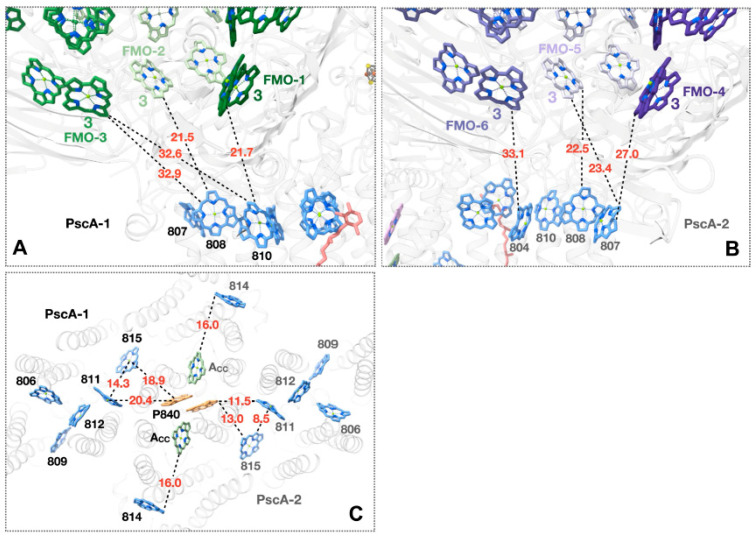
Potential pathways of excitation energy transfer revealed by the structure of the GSB FMO-RC-PS complex [4]. (**A**) Interface region between FMO-1 and PscA-1 subunit at cytoplasmic pigment surface. Interpigment distances (Å) represent BChl edge-to-edge distances. (**B**) Interface region between FMO-4 and PscA-2 subunit showing BChl edge-to-edge distances. (**C**) Potential pathways of energy transfer from nearby light harvesting (LH)-BChls to P_840_ special pair located in the periplasmic bilayer. BChl edge-to-edge distances are shown for PsaA-2-associated BChls, while their Mg-to-Mg distances are shown for their PsaA-1-associated counterparts. Phytyl tails of (B)Chls have been removed for clarity.

**Figure 6 biomolecules-14-00311-f006:**
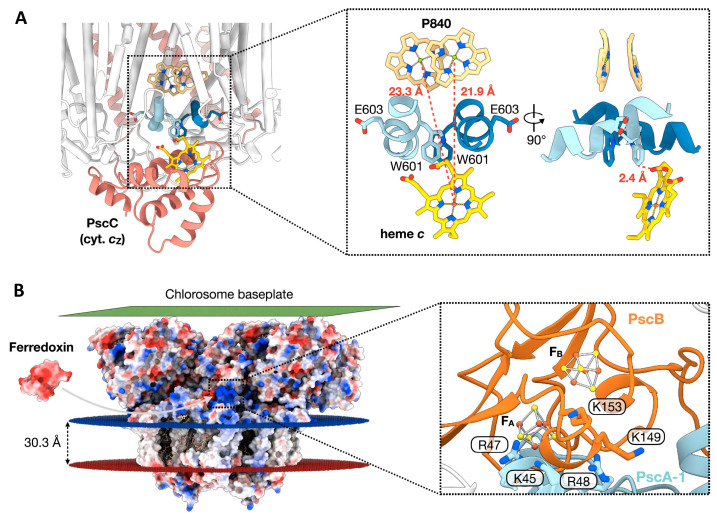
(**A**) Potential *Cba. tepidum* cytochrome *c*_Z_ binding site as predicted from protein–protein docking analysis [4], using the crystal structure of the C-terminal electron carrier domain [41] with enlarged view at right. Red dotted lines in the right panel show respective distances from the heme Fe to the Mg ions of each of the P_840_ special pair BChls. At right, the distance between the heme-associated propionate group and the neighboring PscA Trp601 residues is also shown. (**B**) Potential ferredoxin-PscB binding site [4], shown in the electrostatic surface representation of the *Cab. tepidum* FMO-RC-PS. Protein surface coloring represents the electrostatic surface potential (−10 kT, red; +10 kT, blue). An enlarged detailed view of the putative ferredoxin-PscB docking site is seen at right, along with a number of adjacent positively charged residues provided by both PscA-1 and PscB.

**Figure 7 biomolecules-14-00311-f007:**
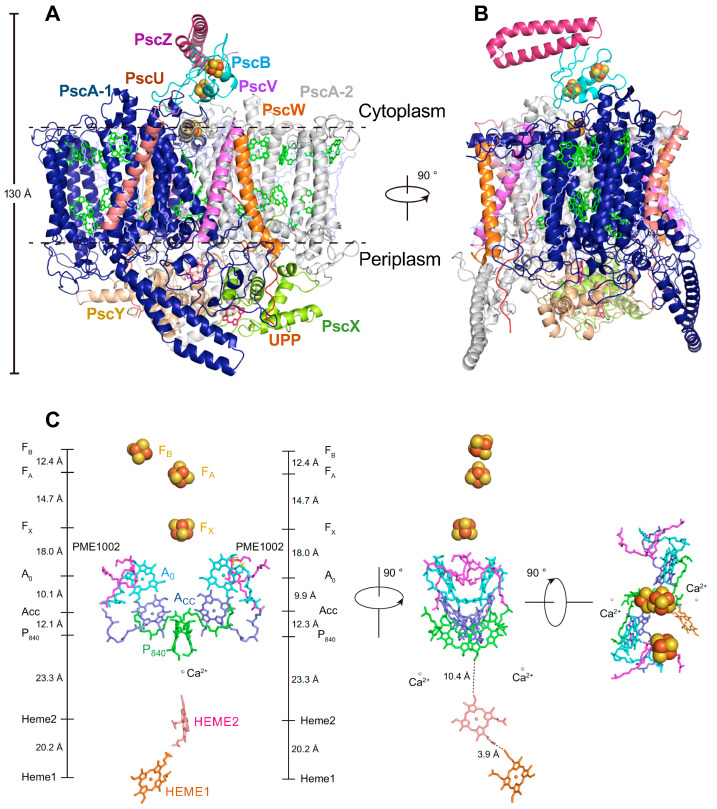
Structure of the Type I homodimeric RC-PS complex of *Cac*. *thermophilum* as determined by cryo-EM at 2.6 Å resolution [6] (PDB ID: 7VZR). (**A**) View parallel to the membrane plane. (**B**) Side view along the membrane plane. Main tetrapyrroles of (B)Chls and Heme groups, green and hot pink, respectively; carotenoids, lipids, and unidentified molecules are shown as line models in light blue; [4Fe-4S] clusters F_X_, F_A,_ and F_B_, are shown as red and yellow spheres. While PscA has a conserved Type I PS arrangement of 11 TMHs, a unique extramembrane loop region protrudes between helices 7 and 8. (**C**) Cofactor arrangement of the *Cab. themophilum* RC electron transfer chain as viewed from membrane plain (**left**), periplasmic surface (**center**), and cytoplasmic surface (**right**). Center-to-center distances between cofactors are presented on the ordinates. PME, phosphatidyl-N-methylethanolamine.

**Figure 8 biomolecules-14-00311-f008:**
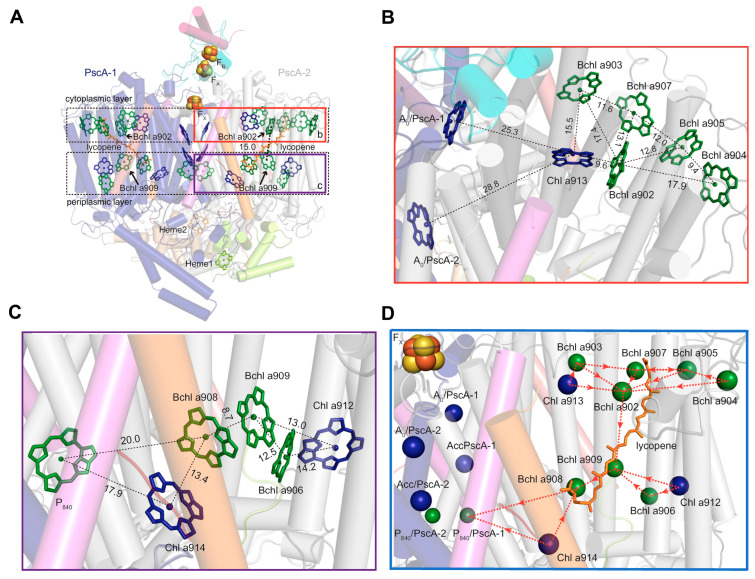
Pigment distribution and proposed excitation energy transfer pathways in the *Cac. thermophilum* RC-PS [6]. (**A**) Layering of pigments in cytoplasmic and periplasmic membrane bilayer leaflets. Chl *a*, dark blue; BChl, green; lycopene, brown. (**B**) Mg^2+^ → Mg^2+^ distances (Å) for PsaA-1 (B)Chls in the cytoplasmic leaflet. (**C**) Mg^2+^ → Mg^2+^ distances (Å) for PsaA-1 (B)Chls in the periplasmic leaflet. (**D**) Proposed excitation energy pathways of PsaA-2 antenna (B)Chl *a* → RC as denoted by the dotted red arrows.

**Figure 9 biomolecules-14-00311-f009:**
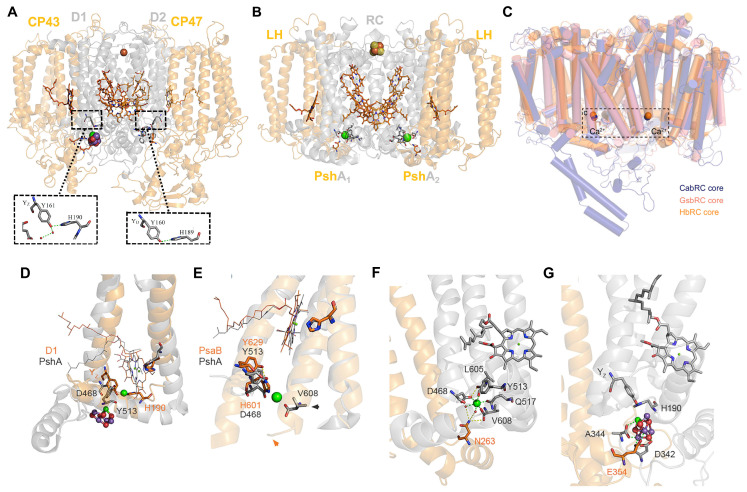
Structural evidence supporting the role of the Ca^2+^-binding site of the HB RC as a possible evolutionary precursor of the PSII H_2_O oxidizing Mn_4_CaO_5_ cluster region [2]. (**A**) PSII RC heterodimeric D1 and D2 RC core polypeptides (gray structures); Chl protein (CP)43 and CP47 antenna proteins (orange structures); cofactors are shown as stick structures. Boxed regions show an enhanced view of homology between the D1 Y_Z_ donor site and Y_D_ of the D2 polypeptide (**B**). The HB (PshA)_2_ homodimeric antenna (orange structures comprising α-helices 1–6) and RC core regions (gray structure comprising α-helices 7–11). Spheres in panels A and B represent Fe (orange), S (yellow), Mn (purple), O (red), and Ca (green) atoms. (**C**) Superpositioning of homodimeric Type I RC Ca^2+^-binding sites within RC-PS structures [6]. (**D**) Enhanced view of conserved regions of TMH 3 and 4 of D1 polypeptide (orange) and helices 9 and 10 of the PshA RC domain (gray). (**E**) Enhanced view of matching regions of PsaB of cyanobacterial PSI RC (orange) and PshA RC (gray), demonstrating that helix 11 is two turns greater in length than PshA counterpart (as demarcated by the black and red arrowheads), as well as the lack of Ca^2+^ binding site in PSI [2]. (**F**) Enhanced view of PshA Ca^2+^-binding site illustrating the connection of ligand of Asn263 of the antenna domain of the PshA protein. (**G**) Enhanced view of the Mn_4_CaO_5_ cluster and Y_Z_ regions of PSII illustrating how the C-terminal residues Glu354 of CP43 and Ala344 of D1 provide direct ligands to the Ca^2+^ atom within the Mn_4_CaO_5_ cluster.

**Figure 10 biomolecules-14-00311-f010:**
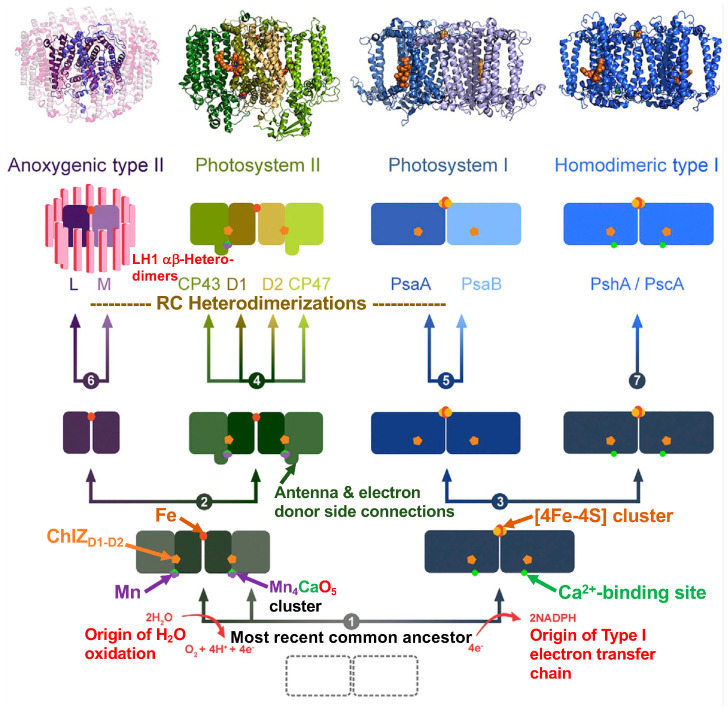
Overview of evolutionary steps leading to the extant Type I and I RCs (adapted from [2]). Step 1 is based upon sequence analyses and structural and functional changes initiated in the common ancestor containing both the Type II and Type I RC precursors. In steps 2 and 3, evolutionary precursors are further separated into distinct RC-PS precursors, ultimately forming the distinct monophyletic Type II and Type I RC clades. Subsequent heterodimerization occurs in steps 4, 5, and 6 to create the respective extant PSII, PSI, and anoxygenic Type II RCs, and then later acquiring the elliptical heterodimeric LH1 antenna, while PSII obtains the CP43 antenna on the D1 side and the CP47 antenna on the D2 side. The arrangement of antenna (B)Chls in PSI and in the homodimeric HB Type II RC-PS is likely to have already been established in their immediate homodimeric precursors. Notably, PSII shares more characteristics with homodimeric Type I RCs than with Type II anoxygenic counterparts, as detailed in the text. The structures shown at the top consist of RC-LH1 complex from *Thermochromatium tepidum* (3.0 Å resolution, Protein Data Bank (PDB) ID: 3WMM), PSII of *Thermosynechococcus vulcanus* (1.9 Å resolution, PDB ID: 3WU2), PSI of *Synecho-coccus elongatus* (2.5 Å resolution, PDB ID: 1JB0), Type I homodimeric RC-PS of *Hmi. modesticaldum* (2.2 Å resolution, PDB ID: 5V8K).

**Table 1 biomolecules-14-00311-t001:** Resolved subunits of the *Cac. thermophilum* reaction center-photosystem structure [6].

Designation	Residues	TMHs ^1^	Functional Assignment
PscA-1	865	11	Homodimeric RC-PS core, LH, and RC charge separation roles
PscA-2	865	11	“
PscB	179	None	Houses terminal RC [4Fe-4S] electron acceptors F_A_ and F_B_
PscU	35	1	Newly identified, unknown function
PscW	45	1	“
PscV	36	1	“
PscX	189	None	Cytochrome *c* electron donor 1 to RC forms heterodimer with PscY
PscY	221	None	Cytochrome *c* electron donor 2 to RC forms heterodimer with PscX
PscZ	70	None	Putative PscB stabilizer may play a PsaD role in enhancing PSI e^−^ transfer
UPP	19	1 ^2^	Unidentified polypeptide, non-helical, unknown function

^1^ Transmembrane α-helices; ^2^ Nonhelical transmembrane span.

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
