# Peer review of "What We Are Learning from the Diverse Structures of the Homodimeric Type I Reaction Center-Photosystems of Anoxygenic Phototropic Bacteria"

_biomolecules, 2024, doi:10.3390/biom14030311_

Round 1
Reviewer 1 Report
Comments and Suggestions for Authors
This is a timely and well-written article that compares and contrasts the solved structures of the homodimeric Type I reaction centers found in heliobacteria, green sulfur bacteria and chloroacidobacteria as well as highlights recent thinking on how homodimeric Type I reaction centers and Photosystem II may (surprisingly) share a common ancestor. In this review, Prof. Niederman has done a masterful job of describing, in detail, these three structures and how the presence of a Ca2+ binding site (as well as the sequences and tertiary structures of the corresponding proteins) has led evolutionary biologists to propose a relationship with the Mn4CaO5 cluster in the D1 subunit of oxygen-producing Photosystem II. These ideas challenge the well-entrenched view that Photosystem II evolved from the anoxygenic bacterial reaction center (leaving the evolutionary origin of the latter unsatisfyingly unclear). This correspondence is not yet widely known outside the photosynthesis community, making this article opportune and relevant. There are no major flaws in the article (although I haven’t checked whether every instance of the numbering of the amino acids is correct), but there are a few minor points that I elaborate below that Prof. Niederman should consider.
First and foremost, the figures are fuzzy when I enlarge the pdf document on my screen. A reader who doesn’t know the field will likely get discouraged in trying to relate what is said in the text with what is shown in the figures. This is especially apparent when there is a lot to analyze as in Figures 3, 6 and 7.
Line 83: substitute ‘transfer’ for ‘transferring’.
Lines 197, 198: The phrase ‘with the final soluble acceptor pool replete’ is not clear
Line 298: On the same topic, there was a lot of controversy in the past as to whether a quinone served as an intermediate acceptor between A0 and FX in both heliobacterial and green sulfur bacterial reaction centers. Recent work has put this idea to rest: there is simply no evidence of a quinone in any of the three structures of the Type I homodimeric reaction centers. However, there is an asymmetry in the review: In lines 297 and 298 Prof. Niederman asks the reader to consider refs. 32 and 33, the former of which argues (incorrectly, as it turns out) for the presence of a quinone in the green sulfur bacterial reaction center, but he doesn’t refer to similar arguments made for the presence of a quinone in the heliobacterial reaction center. My recommendation is this: because this issue is now settled, either delete ‘(see, however [32,33]’ on Line 298 or include a similar ‘(see, however, [X, Y, and Z])’ in the earlier description of the heliobacterial reaction center, where X = Kondo T, Itoh S, Matsuoka M, Azai C, Oh-oka H (2015) Menaquinone as the Secondary Electron Acceptor in the Type I Homodimeric Photosynthetic Reaction Center of Heliobacterium modesticaldum. J Phys Chem B 119:8480-8489; Y = Kondo T, Matsuoka M, Azai C, Kobayashi M, Itoh S, Oh-Oka H (2018) Light-Induced Electron Spin-Polarized (ESP) EPR Signal of the P800(+) Menaquinone(-) Radical Pair State in Oriented Membranes of Heliobacterium modesticaldum: Role/Location of Menaquinone in the Homodimeric Type I Reaction Center. J Phys Chem B 122:2536-2543, and Z = Miyamoto R, Mino H, Kondo T, Itoh S, Oh-Oka H (2008) An Electron Spin-Polarized Signal of the P800(+)A1(Q)(-) State in the Homodimeric Reaction Center Core Complex of Heliobacterium modesticaldum. Biochemistry 47:4386-4393.
Line 328: Please define ‘chlorotrophic’ for readers not familiar with the nomenclature.
Line 396 and Lines 409-411: I think it will be difficult for a reader unfamiliar with the field to understand what is meant by ‘Although both PsdC-1 and PsdC-2 are associated with the GSB, only a single heme c binding site was revealed’. I would suggest that this sentence and the following one be rewritten to clarify the meaning.
Line 736: ‘Cyanobacteria’ should not be capitalized.
Author Response
Reviewer # 1
- The relevant residue numberings have all been checked and abbreviations have been corrected where necessary to three letter designations and they have also been highlighted.
- I apologize about the quality of some of the figures; I submitted high-resolution copies of all the individual figures with the original Ms., but they were apparently not made available to the reviewers.
- Line 83 (now 88): substitute ‘transfer’ for ‘transferring.’ Done
- Lines 197,198 (now 205-208): The phrase ‘with the final soluble acceptor pool replete’ is not clear.
Response: I have hopefully clarified the role of MQ under high-light conditions by adding that it is a lipophilic molecule able to function in an analogous manner to the mobile quinones in Type II RCs. A sentence has also been added to further clarify the MQ role: “It is thought that under these circumstances, electrons are instead transferred directly to the cytochrome b6c complex, rather than through the ferrodoxin:NADP+ oxidoreductase and NAD(P)H dehyrogenase as is the case with ferredoxin.”
- Line 298 (now 304): On the same topic……
Response: I have eliminated “see however [32,33]” and replaced this with “see however [32-34]” and included the work of the Oh-Ota group as the new refs. 32-34 as you have suggested.
- Line 328 (now 336): Please define ‘chlorotrophic’ for readers not familiar with the nomenclature.
Response: ‘Chlorotrophic’ here was a misspelling – meant to be chlorophototrophic. Chlorophototrophic first appears in line 69-70: “in the well-studied anoxygenic purple chlorophototrophic bacteria,” has now been revised to read: in the well-studied anoxygenic purple chlorophototrophic bacteria (i.e., capable of forming chlorophylls).
- Line 396 (now Lines 410-411): I think it will be difficult for a reader unfamiliar with the field to understand what is meant by ‘Although both Psd PscC-1 and Psd PscC-2 are associated with the GSB, only a single heme c binding site was revealed’. I would suggest that this sentence and the following one be rewritten to clarify the meaning.
Response: I have extensively revised this section (now Lines 410-411; 425-428) which now reads: “Although both the cytochromes cZ proteins (PscC-1 and PscC-2) are associated with the PscA homodimer, only a single heme c binding site was revealed [2], similar to the situation in the cyanobacterial PSI [41] and the RC of purple bacteria [42]. The parallel PscA-1 and PscA-2 Trp residues (Trp601) are associated at the docking interface with heme c, located in the PscC crevice, and are positioned at a distance of 2.4 Å from the heme c propionate group (Figure 6A).
- Line 736: ‘Cyanobacteria’ should not be capitalized. Fixed, now Line 781.
- Response: All mentions of cyanobacteria are now in lower case
Reviewer 2 Report
Comments and Suggestions for Authors
The author’s contribution to deep understanding photosynthetic apparatus in anoxyphotobacteria has been acknowleged worldwide. This time, Professor Robert A. Niederman delivers a very informative and timely report about gene determinants, molecular structure, and functional behavior of Type-I reaction centers in the heliobacteria, green sulfur bacteria, and green acidobacteria. The manuscript gives a comprehensive picture of current knowledge on this sophisticated bioenergy vehicle. Thus, with some minor changes, the review article submitted by Professor R.A. Niederman will suit for publication in Biomolecules.
General remarks
The quality of manuscript is very high in nearly all respects. Nevertheless, my first recommendation is to introduce a separate “Conclusion” (Section 4) which could be somehow based on L29-L45, in the Abstract, and L733-L749, in the Main text. My second recommendation is to supply the article with a schematic picture demonstrating hypothetical evolution of RCs, including scenario(s) for WOX complex origin. Finally, I recommend the author paying more attention to text formatting.
Minor remarks
The text should be carefully checked throughout:
L18: chlorophyll
L33 and thereafter: Mn4CaO5
L65 and thereafter: remove the strange term “chlorophototrophic”
L69: the abbreviation CAB is not good because it commonly denotes light-harvesting chlorophyll a/b proteins; what about using the abbreviation CB instead
L74: delete “bacterial”
L87: separation
L108: see comment for L69
LL1089: respective genes
L117: pigment proteins
L124: phylum
L131 and thereafter: Earth
L145: modesticaldum
L147-158: figure caption should not occupy positions at separate pages
L166: the footnote should be marked with an asterisk
L172 and thereafter: (PshA)2
L212: represent
L222: molecules participating in
L250: dodecyl
L273, 276, 277: kDa
L289: in both sides
L328: remove the strange term “chlorotrophic”
L390: delete the comma
L428: pscD gene
L448: delete “prokaryotic”
L459: cryo-EM
L484-388 (Table 1): replace capital letters in the table heading, and correct table format
L490 and thereafter: H2O
L494: see the above comment, and use letter “O” instead of number “0”
L501: use a point and a gap after “leaflet”
L502: use a gap before (D)
L530: O2
L549: delete “chlorophototrophic”
L565: puf operon
L572: replace “the filamentous anoxygenic phototroph” with “chlorosome-lacking green non-sulfur bacterium”
L575: delete “a green nonsulfur bacterium lacking chlorosomes”
L624: A0 cofactor (number “0”, not letter “O”)
L658: use numbers for PshA and PscA in subscript
L700: Mn2+ cations
Ref1: Fromme, P. (Petra)
Ref3: remove a semicolon before “et al.”
Ref5: add a comma before “USA”
Ref14: Rhodopseudomonas viridis
Ref15: add a gap before journal number
Ref20: add a gap after “Gorka,”
Ref25: delete semicolons before and after the initials “L.E.”
Ref26: in the case of Merkel, replace a comma between the initials with a point
Ref31: Oh-Oka
Ref33: remove “DE”
Ref39: Oh-Oka
Ref40: Chlorobium tepidum
Ref43: add a comma after «Kurisu”
Ref45: use points instead of commas between the initials of Costas, Maresca, and Chew
Ref48: Candidatus Chloracidobacterium thermophilum (see original title)
Ref49: Candidatus Chloracidobacterium thermophilum (see original title)
Ref51: add a semicolon after “Cannife, D.P.”
Ref57: delete a semicolon before “et al.”
Ref65: Beatty, J.T.
Ref68: add a gap after “2004”
Ref70: add commas after “Cardona” and “Sanchez-Barakaldo”
Author Response
Reviewer # 2
- In response to the first recommendation “to introduce a separate Conclusions section,” this can now be found on Lines 846-903 of revised Ms. and includes the suggested material.
- In response to the second recommendation “to supply the article with a schematic picture demonstrating hypothetical evolution of RCs, including scenario(s) for WOX complex origin,” this can now be found as on Line 782 as Figure 10: Overview of evolutionary steps leading to the extant Type I and I RCs.
- In response to the third recommendation that more attention should be paid to text formatting especially as seen in the Figure 1 legend which appears on split pages pp. 3-4, it has been dealt with by changing positions of all Figs. and legends to conform.
- Minor remarks: All suggested text corrections were made except the following:
L 65: I have retained the term “chlorophototrophic” which is now defined on L. 70 of revised Ms.: (i.e., capable of forming chlorophylls) which importantly distinguishes anoxygenic purple phototrophic bacteria from purple halophiles that carry out bacteriorhodopsin catalyzed light transduction.
- L. 572,575 (now 609-616): I have retained the description of castenholzii in separate sentences which I believe now reads better.
Ref. 1: Author is Raimond Fromme, not Petra
Reviewer 3 Report
Comments and Suggestions for Authors
Robert A. Niederman provided a detailed analysis of the recently revealed structures of the homodimeric Type I RC-PSs from HB, GSB, and CAB. The focus was on their structural variations and how they contribute to a better comprehension of the LH and primary photochemical processes facilitated by these distinctive pigment-protein complexes. Furthermore, author explained how these structures have contributed to our current knowledge of the evolutionary origins of RC-PSs, particularly in relation to the relationships that the observed homodimeric architectures have with the development of the existing heterodimeric PSI and PSII complexes found in oxygenic phototrophs..
Throughout the content, the authors have successfully encapsulated the dynamic interplay between modern technologies and scientific innovation, underscoring its potential to revolutionize the field, making it more efficient and accessible. This article promises to offer valuable insights into the rapid developments in structural biology, emphasizing the importance of evolutionary origins of these molecules.
Given the clarity, depth, and relevance of the content presented, I believe this article is well-positioned to contribute significantly to the current literature in the field. It will undoubtedly provide readers with a comprehensive understanding of the current state-of-the-art methods.
In general, the paper exhibits a commendable level of writing proficiency, and the examination of state of the art methods in developing therapeutic proteins is intellectually stimulating. The authors' inclusion of fundamental concepts in initial section is deemed significant and contributes to the extent body of literature. Tables presented in the paper constitutes a significant inclusion that has the potential to enhance the paper's citation count.
Author Response
Reviewer # 3
Thank you for your very positive response.